**Data Availability Statement:** The data that support the findings of this paper are available upon reasonable request and with the approval of

# Experiences and lessons learned from community-engaged recruitment for the South Asian breast cancer study in New Jersey during the COVID-19 pandemic

Jaya M. Satagopan[1]*, Tina Dharamdasani[1], Shailja Mathur[2], Racquel E. Kohler[3], Elisa V. Bandera[4], Anita Y. Kinney[1]

1 Department of Biostatistics and Epidemiology, School of Public Health, Cancer Institute of New Jersey, Rutgers, The State University of New Jersey, New Brunswick, NJ, United States of America, 2 Department of Family and Community Health Sciences, Cooperative Extension, Rutgers, The State University of New Jersey, New Brunswick, NJ, United States of America, 3 Department of Health Behavior, Society & Policy, School of Public Health, Cancer Institute of New Jersey, Rutgers, The State University of New Jersey, New Brunswick, NJ, United States of America, 4 Cancer Epidemiology and Health Outcomes, Cancer Institute of New Jersey, Rutgers, The State University of New Jersey, New Brunswick, NJ, United States of America

* satagopj@sph.rutgers.edu

## Abstract

### Background

South Asians are a rapidly growing population in the United States. Breast cancer is a major concern among South Asian American women, who are an understudied population. We established the South Asian Breast Cancer (SABCa) study in New Jersey during early 2020 to gain insights into their breast cancer-related health attitudes. Shortly after we started planning for the study, the COVID-19 disease spread throughout the world. In this paper, we describe our experiences and lessons learned from recruiting study participants by partnering with New Jersey's community organizations during the COVID-19 pandemic.

### Methods

We used a cross-sectional design. We contacted 12 community organizations and 7 (58%) disseminated our study information. However, these organizations became considerably busy with pandemic-related needs. Therefore, we had to pivot to alternative recruitment strategies through community radio, Rutgers Cancer Institute of New Jersey's Community Outreach and Engagement Program, and Rutgers Cooperative Extension's community health programs. We recruited participants through these alternative strategies, obtained written informed consent, and collected demographic information using a structured survey.

### Results

Twenty five women expressed interest in the study, of which 22 (88%) participated. Nine (41%) participants learned about the study through the radio, 5 (23%) through these participants, 1 (4.5%) through a non-radio community organization, and 7 (32%) through

Rutgers University Institutional Review Board (IRB Pro2020002217). The data are not publicly available as it contains information that could compromise the privacy of research participants. Given that participants are women of South Asian origin living in the US, and the dataset contains their county of residence and age, making the data available can risk the possibility of participants being identified. Please see more information here: https://policies.rutgers.edu/10011-currentpdf. Please contact the lead author (Jaya M. Satagopan; satagopj@sph.rutgers.edu) as well as Rutgers University's Institutional Review Board (IRBOffice@research.rutgers.edu) with any request for data access.

**Funding:** The authors received no specific funding for this work.

**Competing interests:** The authors have declared that no competing interests exist.

community health programs. Two (9%) participants heard about the study from their spouse. All participants were born outside the US, their average age was 52.4 years (range: 39–72 years), and they have lived in the US for an average of 26 years (range: 5–51 years).

## Conclusion

Pivoting to alternative strategies were crucial for successful recruitment. Findings suggest the significant potential of broadcast media for community-based recruitment. Family dynamics and the community's trust in our partners also encouraged participation. Such strategies must be considered when working with understudied populations.

## Introduction

South Asians are individuals tracing their ancestry to Bangladesh, Bhutan, India, Maldives, Nepal, Pakistan, and Sri Lanka [1]. South Asians are among the fastest growing racial/ethnic minorities in the United States (US), increasing by over 46% from around 3.86 million in 2010 to around 5.6 million in 2019 [2,3]. New Jersey (NJ) is one of the most diverse states in the US where over 940,000 (10%) of the state's nearly 9.3 million residents are Asians [4]. More than 44% of NJ's Asians are South Asians, making them the largest Asian subgroup in the state. South Asians living in NJ and elsewhere in the US, henceforth referred to as South Asian Americans, confront a wide range of health issues and are an at-risk population for disproportionate burden of chronic diseases, including cancer [5–7].

Despite their growing numbers, South Asian Americans remain one of the most understudied minority populations in health research [5,8,9]. The paucity of data on the health of South Asian Americans has been attributed to aggregation of Asian American subgroups under a combined "Asian American and Pacific Islander" category in most studies and to lack of participation and engagement of the South Asian American community in scientific studies of breast cancer [5,9–11]. Although large population-level health studies (for example, the National Health Interview Survey) now obtain disaggregated race/ethnicity data, the sample sizes of Asians and Asian subgroups in these data sets remain small and few research investigations of these large studies report results disaggregated by Asian American subgroups [12].

Breast cancer is a growing concern among South Asian American women due to their rapidly increasing breast cancer incidence [13–16]. Compared to non-Hispanic White women, South Asian American women are more likely to be diagnosed with breast cancer at younger ages and with advanced disease [17–20]. However, there is limited breast cancer-related data available from South Asian American women to understand these disparities and little is known about their breast cancer-related health attitudes and experiences [21,22]. Breast cancer is amenable to risk reduction, screening, and early detection. Yet, studies have reported that South Asian Americans have low mammogram screening rates [23–26]. Research investigations into breast and other cancer-related health have called for further studies to understand the determinants of adherence to age-appropriate screening behaviors in South Asian American women [27,28]. Qualitative studies of South Asians from the United Kingdom (UK) and Canada have reported spiritual connection, fear about cancer, and pain as barriers to cancer screening, and a desire to be healthy to be a motivator for screening [29–33]. A qualitative study of mammography screening practices of Muslim American women, which included Muslim women of South Asian origin, reported that theocentric views of cure and illness inform health decision frames that impact mammography intention [33]. Despite such

investigations, a comprehensive understanding of breast cancer-related health attitudes of South Asian American women remains low.

Therefore, we established the South Asian Breast Cancer (SABCa) study at Rutgers, The State University of New Jersey (Rutgers University), the home institution of the authors of this study. Plans for the study began during early 2020 with the overarching goal of gaining insights into breast cancer-related health attitudes and healthcare experiences of South Asian American women. This study consisted of 3 components. The goal of Component 1 was to recruit South Asian American women without a prior diagnosis of cancer by partnering with community organizations and conduct focus group discussions to assess knowledge, attitudes and perceptions about breast cancer, risk factors, preventive care, and participation in breast cancer-related scientific studies among these women. The goal of Component 2 was to recruit South Asian American women diagnosed with breast cancer by partnering with the NJ State Cancer Registry and conduct focus group discussions to assess knowledge, attitudes and perceptions about survivorship care, follow-up care, and participation in breast cancer-related scientific studies among these women. The goal of Component 3 was to recruit healthcare professionals from Rutgers University's hospitals and conduct structured interviews to assess their experiences in providing care to South Asian Americans for breast cancer or any chronic disease.

In this paper, we focus on community-based recruitment of South Asian American women to Component 1 focus groups. We describe our community-based recruitment approaches. We did not plan community-based recruitment for Components 2 and 3, for which we planned partnership with Rutgers Hospitals to recruit a healthcare professionals and partnership with NJ State Cancer Registry to recruit South Asian American women with breast cancer, both of which were ongoing at the time of preparing this manuscript. Analysis and results of the focus groups discussions for Component 1, recruitment, and analysis of Components 2 and 3, and cross-component comparisons are outside the scope of this paper and will be reported elsewhere.

According to the US Census, New Jersey has the third largest South Asian American population in the US after California (CA) and Texas [34]. Over 9.1% of South Asians in the US live in NJ. Around 5% of NJ's population are of South Asian origin. This concentration of South Asian American community supports the feasibility for the SABCa study to recruit participants through community partnerships in NJ.

Several studies have recommended various strategies such as engagement with community leaders and community-based organizations, targeted communications on social forums, and chain referral approach for recruitment of South Asians in the community [5,6,35]. If our study were to be conducted prior to the COVID-19 pandemic, recruitment methods would typically include disseminating study information by attending community venues (such as places of worship and community centers) and in-person community-based events (such as religious events, cultural events, and health events). We reached out to various South Asian community organizations in NJ to partner with them to disseminate information about the SABCa study and to recruit participants. However, the COVID-19 pandemic began shortly after we established our study. The pandemic introduced unique challenges for community-based recruitment. The South Asian community organizations in NJ suspended their in-person meetings and events due to lock downs and social distancing requirements [36]. Further, many organizations were called to develop and implement pandemic-related responses for the communities they serve, limiting their ability to assist with community-based recruitments for our study [37,38]. Additionally, due to the devastating delta variant wave in South Asia between March and May 2021, much of the South Asian American community was engaged in providing emotional support to their families living in South Asia [39,40], which curtailed community-based recruitment efforts during this period.

People were spending considerable time on the internet during the pandemic, and social media platforms were increasingly used for transnational communications by the South Asian diaspora. Advertising the study through social media platforms was a potential possibility. However, the pandemic also became a period of infodemic when misinformation and low-credibility information were spreading rapidly through social media platforms [41,42]. Studies were also reporting on the adverse impact of social media use on symptoms of fear, anxiety, and depression during this time [43,44]. Therefore, we had to be wary of placing advertisements about our study on social media. We did not want the advertisement to be lost in the vast volume of infodemic. However, people were increasingly engaging with social media of trusted sources such as community organizations with which they had previously established relationships. Therefore, we encouraged the community organizations to share our study information through their social media platforms. However, due to considerable burden on the time and resource of community organizations during the pandemic, we had to identify alternative nuanced approaches to recruit participants from NJ's South Asian community.

This paper summarizes the approaches that we pursued for community-based recruitment during the pandemic to recruit South Asian women without a prior diagnosis of breast cancer to the SABCa study. We reflect on our experiences and summarize our lessons learned. Our experiences and lessons learned offer effective strategies for partnering with the South Asian American community in the future even beyond the pandemic and can be useful for establishing successful collaborations with other understudied populations as well.

## Materials and methods

This is a cross-sectional design, where we collected data from study participants at a single point in time [42,45,46]. The study population consisted of South Asian American women. Planning for recruitment began in March 2021. Recruitment was done locally in NJ between June 2021 and July 2022 by partnering with various organizations serving the South Asian American community. The Rutgers University Institutional Review Board (IRB) approved the study. All study participants provided electronic written informed consent through a Rutgers-approved secure Docusign® platform.

### Eligibility criteria

Eligibility criteria included: (1) self-identified as having South Asian ethnic or cultural ancestry; (2) self-identified as female; (3) age 25 years or above; (4) self-reported as having no prior history of any cancer; (5) able to read and speak English; (6) able to understand and be willing to provide informed consent; and (7) have access to an electronic device (such as computer or tablet or smart phone) to provide electronic consent and to attend an online focus group session. The last criterion was precipitated by the COVID-19 pandemic when in-person meetings were suspended for an extended period.

### Target sample size

In qualitative studies that use focus groups, data saturation is the gold standard approach for determining the number of focus groups and, thus, the total number of individuals in a study. By definition, saturation can be determined only during or after data analysis [47]. Research and budgetary planning require an initial guidance on the sample size before the study is implemented [47]. This was especially crucial during the COVID-19 pandemic when Rutgers University implemented a spending freeze, warranting forethought on a target sample size for budgetary planning of our study [48].

Guest and colleagues conducted an empirical study of 40 focus groups to derive non-probability sample sizes guidelines and found that 90% of all themes in a study are discoverable within 2 to 6 focus groups, 3 focus groups are enough to identify the most prevalent themes, and theme discovery reaches a plateau at 4 focus groups [47]. Hennink and colleagues also found that 4 focus groups are sufficient to achieve saturation [49]. Krueger and Casey recommend 5 to 8 participants per focus group [50]. Building on these guidelines, we planned our study budget to recruit 18 to 32 eligible participants to conduct 3 to 4 focus groups with around 6 to 8 participants per focus group.

## Recruitment methods

We began planning the SABCa study in 2020. After obtaining IRB and budgetary approval, we began our recruitment efforts during early March 2021 and recruited participants between June 2021 and July 2022. We used several approaches for recruitment: partnering with community organizations who disseminated study information to the community they serve, describing the study at a community-focused Science Café, disseminating study information through community-focused health education, and snowball sampling. Convenience based sampling was used for all recruitment methods where interested participants would contact study staff to enroll in the study. Table 1 shows the chronology of our recruitment efforts, which we describe below. Study outreach efforts were done in English language only since we did not have the resources to translate study materials to multiple South Asian languages or to recruit research staff with multiple language skills during the COVID-19 pandemic.

**Study flyer.** We prepared our study flyer by incorporating specific aspects of South Asian culture. The flyer had a watermark showing heaps of spices that are commonly used in the South Asian culture. The flyer was largely yellow in color to represent turmeric, a commonly used spice. The flyer border and the horizontal lines to separate information were in red color

**Table 1. Timeline of community engagement activities.**

| TIME | COMMUNITY ENGAGEMENT ACTIVITY | |
|---|---|---|
| June 2020 | Began interactions with Rutgers Co-op. PI invited by Rutgers Co-op to discuss breast cancer in radio talk show. | |
| | | |
| March 2021 | Introduced study to South Asian cultural, religious, and health screening organizations in NJ. Presented study at CINJ COE's Community Science Café. | Sustained communications with South Asian community in NJ. |
| April 2021 | | |
| May 2021 | Advertised study through radio. Participated in radio talk show about breast cancer. Introduced study to health service organizations in NJ. Participant enrollment initiated. | |
| June 2021 | | |
| July 2021 | Described our research program and study in Citizen Science sidewalk science video. | |
| | | |
| October 2021 | Enrollment crossed 10 participants. | |
| | | |
| | | |
| May 2022 | Presented study at Rutgers Co-op's community-focused online health education program. | |
| July 2022 | Enrollment reached 22 participants. | |
| | | |
| September 2022 | Described our research program and study at a South Asian community event in NJ. | |
| October 2022 | Wrote article about breast cancer for a South Asian community magazine in NJ. | |

Abbreviations: Rutgers Co-op = Rutgers Cooperative Extension, CINJ = Rutgers Cancer Institute of New Jersey, COE = Community Outreach and Engagement office.

to represent powdered chili, a commonly used spice, and to represent henna and vermillion, which are commonly used for cosmetic needs in the South Asian culture (see S1 Fig).

**South Asian community-based organizations.** During early March 2021, we contacted various South Asian cultural, religious, and health screening organizations in NJ. We identified these organizations through our previously established strong community connections: Two of this paper's authors (JMS and SM), who are of South Asian origin and have lived in NJ for over 25 years, have long-standing social interactions with these cultural and religious organizations. South Asian health screening organizations offer free services such as blood test, eye screening, dental screening, physical examination, cardiology evaluation, physical therapy, weight management counseling, and mental health counseling, amongst others, to the South Asian community. We also presented our study at a Community Science Café organized by Rutgers Cancer Institute of New Jersey Community Outreach and Engagement (CINJ COE) office, where community members bring their unique perspectives on research to provide input to scientists. Upon recommendation from the Community Science Café audience, we contacted health service organizations in NJ whose catchment population include the South Asian community. These health service organizations provide, finance, supervise and evaluate health care activities in NJ.

We introduced the SABCa study and explained the eligibility criteria to these cultural, religious, health screening, and health service organizations and requested them to share the study flyer with their community listserv via e-mail. While we did not place advertisements about our study in social media due to infodemic concerns, we encouraged the community organizations to share the study information through their social media groups. Since these organizations are trusted by the South Asian American community, we felt that community members are more likely to engage with the study flyers shared by the organizations through their social media groups without dismissing it as a COVID-19-related infodemic. However, we did not track the social media pages of these organizations to check engagement of community members with advertisements placed by community organizations in their social media pages. The COVID-19 pandemic was still in full swing during March 2021. We reached out to a member of the leadership team (such as President or Secretary) of 12 community organizations, of which 7 (58%) agreed to disseminate the study flyer. However, these community organizations quickly became busy assisting their communities with pandemic-related needs and were unable to disseminate the study information frequently or widely as originally planned. Only one participant who heard about the study through community organizations contacted the SABCa study staff expressing interest in the study.

Within two weeks of sending our study flyers to the community organizations, South Asia, especially India, experienced an intense delta variant pandemic wave. This wave began in mid-March and lasted for around 3 months and declined after peaking in mid-May 2021 [51]. Several South Asian Americans were providing emotional support to their families living in South Asia. To empathize with the South Asian American community, we paused recruitment during this period by not repeatedly requesting the community organizations to disseminate study materials. We used this period to learn more about South Asian Americans' sources of credible information during this challenging time. Discussions amongst the study team and conversations with personal contacts revealed that broadcast media (radio) catering to South Asian Americans and Rutgers-based community engagement programs are likely to be constructive recruitment strategies during this time.

**Broadcast media (radio).** During the COVID-19 pandemic, public media consumption started increasing and many were listening to as much or more radio compared to before the pandemic [52]. Radio personalities such as radio jockeys and talk show hosts were providing timely information about the pandemic and became people's primary connection to the

outside world during a time of considerable uncertainty and disrupted routines. The public increasingly held radio personalities in high regard and trust [52]. There are numerous radio stations in New Jersey. One of these radio stations is widely heard in New Jersey, New York, and Eastern Pennsylvania, reaching a population of over 500,000 in the listening areas. This radio station serves the South Asian community by offering a unique blend of programs including popular South Asian Bollywood film music, news programs, and talk shows about South Asian recipes, spirituality, society, and health, amongst other topics. The radio personalities cater to all South Asian ethnicities, transcend age groups and boundaries, and expose listeners to South Asian culture. Thus, this radio station, which we refer to henceforth as community radio station, is popular among South Asians living in NJ. Rutgers Cooperative Extension (Rutgers Co-op), which provides a diverse range of research, extension, and education programs to the people of NJ, partners regularly with this community radio station to disseminate health education about chronic diseases for the community. During the planning stages of the SABCa study in 2020, Rutgers Co-op invited the PI for a discussion about breast cancer in a radio talk show during June 2020. Leveraging these prior interactions, we partnered with this community radio station during May 2021 and promoted our study through their commercial time slots between late-May 2021 and late-June 2021 (see S2 Fig).

The radio personalities read the script with South Asian music playing in the background during short breaks in popular Bollywood film music programs, news, or talk shows, once in the morning and once in the evening. They began narrating the script by saying "namaste", which means "hello" in several South Asian languages and closed by saying "dhayavad" or "shukriya", which means "thank you". The community radio station also invited us to participate in a talk show about breast cancer during June 2021. The PI described the SABCa study in this talk show program and encouraged listeners to share the information in their community networks. A total of 11 individuals contacted us expressing interest after hearing about the study through the radio.

**Rutgers-based community outreach programs.**   We worked closely with two Rutgers-based community outreach programs–Rutgers Cancer Institute of New Jersey Community Outreach and Engagement office (CINJ COE) and Rutgers Cooperative Extension (Rutgers Co-op). During July 2021, we described our breast cancer research program, including this study, in a Science Café sidewalk video prepared and disseminated by CINJ COE. The Science Café sidewalk video is a community-focused short video where a researcher summarizes their research program, its significance, and impact to community members. The PI summarized the SABCa study in a Science Café sidewalk video. The CINJ COE office distributed the video to the entire NJ community through social media–especially, Facebook and LinkedIn. Rutgers Co-op conducts multiple online synchronous health programs for South Asian Americans living with chronic diseases. We presented our study at a Rutgers Co-op online community health program during May 2022. We distributed the study flyer through this program and encouraged program participants to share the information through their networks. We also shared the study flyer widely through Rutgers Co-op's community networks. Eight individuals heard about the study through Rutgers Co-op and contacted us expressing interest in the study.

**Snowball sampling.**   Previous studies have successfully recruited participants using snowball sampling strategy and recommend using this approach for recruiting hard-to-reach populations such as South Asian Americans [6] and Vietnamese Americans [12]. Therefore, we encouraged the participants recruited through different approaches described above to share the study information with their network and refer their eligible and interested contacts to our study. Five acquaintances of the individuals who heard about our study through radio promotions contacted us expressing interest in the study.

**Participant enrollment.** Interested individuals contacted study staff using the IRB-approved contact information provided in the study flyer or in the radio script. Two authors (JMS and TD), who are women of South Asian origin, served as the study staff. Some individuals who heard about the study through Rutgers Co-op initially contacted one of the authors (SM), who is a Rutgers Co-op staff and a woman of South Asian origin. Thus, there was cultural concordance between study contacts and study participants. One of the authors (SM), who is widely known in NJ's South Asian community, explained the study to individuals contacting her, and then forwarded the contact information of eligible interested individuals to the study staff. The study staff explained the study to each interested individual, answered their questions, confirmed eligibility, and registered them by taking down their name and contact information. The staff sent participants a sample electronic copy of the informed consent document and a link to a Rutgers-approved online virtual room to complete the informed consent and a structured survey at a mutually agreed date and time. This allowed the participants to review the informed consent in advance and contact the study staff with any questions.

**Participant survey.** To understand the characteristics of the participants recruited through our approaches, we used a structured survey hosted on a Rutgers-approved secure Qualtrics® online survey platform. The survey was administered in English language. We obtained information on **sociodemographic characteristics** such as participants' age, languages spoken, highest education level, marital status, and country of birth, and **acculturation-related characteristics** such as number of years lived in the US, languages spoken with friends, and type of engagement in the South Asian community. We offered an incentive of $25 in electronic gift card to each participant for completing the survey.

**Procedure for completing informed consent and survey.** Study staff met with each participant in a Rutgers-approved secure virtual room to complete the informed consent and the survey. Participants signed informed consent via a Rutgers-approved online DocuSign® electronic agreement platform. Study staff guided participants with any technology questions that arose when using this platform.

## Statistical analyses

We calculated "cooperation rate" as the ratio of the number of women participating in the study to the number who called the study phone number expressing interest in the study. We calculated descriptive statistics for the survey responses and tabulated them overall and according to recruitment approach i.e., through community organizations or radio, partnership with Rutgers Co-op, or snowball sampling. We conducted these analyses using the SAS software version 9.4 (SAS Institute, Cary, NC). We did not conduct formal hypothesis tests to compare participant characteristics across the recruitment methods due to limited sample size.

## Results

A total of 25 women of South Asian origin contacted us expressing interest in our study. Three (12%) women– 2 that heard about the study from radio promotions and one that heard through Rutgers Co-op–did not respond to subsequent communications from the study staff. After 5 failed attempts to reach them, we assumed that they were no longer interested in the study. All the remaining 22 women satisfied the eligibility criteria and completed informed consent and the survey. We did not lose any participant once they provided consent. Therefore, our total sample size was 22 participants. The overall cooperation rate was 88% (= 22/25), and the rates were 82% (= 9/11) for recruitment through community radio, 87.5% (= 7/8) for community-focused health programs, and 100% (= 5/5) for snowball recruitment. Of the 22 participants, 10 (45%) were recruited through community organizations or radio, 7 (32%)

were recruited through Rutgers Co-op, and 5 (23%) were recruited through snowball sampling.

Table 2 summarizes the participant characteristics. The average age based on all 22 participants was 52.4 years (range: 39 to 72 years) and the average number of years they have lived in the US was 26 years (range: 5 to 51 years). The average age was at least 50 years and the average number of years lived in the US was at least 22 years in each recruitment approach–community organization or community-based radio, Rutgers Co-op, and snowball sampling. All the participants were born outside the US– 95% of them were born in India (other countries of birth not given to protect confidentiality of participants due to small sample size). All the participants recruited through community organizations or community-based radio, 57% of those recruited through Rutgers Co-op, and 80% of those recruited through snowball sampling reported speaking at least two South Asian languages. At least 60% of the participants used English and South Asian languages equally or English more than South Asian languages to speak with friends or used English as a preferred language for movie or television or radio programs, regardless of the recruitment approach. Most participants had master's degree or professional degree or doctoral degree (60% recruited through community organizations or

**Table 2. Characteristics of study participants.**

| Sample Characteristics | Community organization and community-based radio | Rutgers Co-op | Snowball | Total Sample |
|---|---|---|---|---|
| **Total number of participants** | 10 (45%) | 7 (32%) | 5 (23%) | 22 |
| **Age: Mean (SD) (Range)** | 53.2 (7.3) (Min: 39, Max:64) | 53.1 (7.2) (Min: 43, Max: 66) | 50.0 (12.7) (Min: 41, Max: 72) | 52.4 (8.4) (Min: 39, Max: 72) |
| **Number of years lived in the U.S. Mean (SD) (Range)** | 29.0 (9.4) (Min: 19, Max: 50) | 24.1 (10.8) (Min: 5.5, Max: 35.0) | 22.4 (17.3) (Min: 5, Max: 51) | 26.0 (11.7) (Min: 5, Max: 51) |
| **Languages spoken*** | | | | |
| **English** | 10 (100%) | 7 (100%) | 5 (100%) | 22 (100%) |
| **Gujarati** | 6 (60%) | 1 (14%) | 1 (20%) | 8 (36%) |
| **Hindi** | 9 (90%) | 7 (100%) | 4 (80%) | 20 (91%) |
| **Tamil** | 3 (30%) | 0 (0%) | 3 (60%) | 6 (27%) |
| **Number of South Asian languages spoken** | | | | |
| **1** | 0 | 3 (43%) | 1 (20%) | 4 (18%) |
| **2–5** | 10 (100%) | 4 (57%) | 4 (80%) | 18 (82%) |
| **Language spoken with friends** | | | | |
| **South Asian language only or better than English** | 0 (0%) | 1 (14%) | 2 (40%) | 3 (14%) |
| **South Asian and English equally** | 4 (40%) | 5 (72%) | 3 (60%) | 12 (54%) |
| **English only or better than South Asian language** | 6 (60%) | 1 (14%) | 0 (0%) | 7 (32%) |
| **Preferred language for movies, television, or radio programs** | | | | |
| **South Asian language only or better than English** | 1 (10%) | 2 (29%) | 2 (40%) | 5 (23%) |
| **South Asian and English equally** | 4 (40%) | 4 (57%) | 2 (40%) | 10 (45%) |
| **English only or better than South Asian language** | 5 (50%) | 1 (14%) | 1 (20%) | 7 (32%) |
| **Education** | | | | |
| **College or some college** | 4 (40%) | 3 (43%) | 0 (0%) | 7 (32%) |
| **Master's degree or Professional or Doctoral degree** | 6 (60%) | 4 (57%) | 5 (100%) | 15 (68%) |

*The total number exceeds 22 since some participants spoke multiple South Asian languages. Other languages spoken by participants (not shown in the table due to small frequencies): Marathi, Kutchi, Urdu, Malayalam, Punjabi, Telugu. Abbreviation: SD = standard deviation, Min = minimum, Max = maximum.

community-based radio, 57% recruited through Rutgers Co-op, and 100% recruited through snowball sampling). All the participants reported being actively engaged in New Jersey's South Asian community through religious organizations, cultural events for children or adults, social events with friends or family, and volunteering opportunities for the community.

Most of the study participants were from Middlesex County (41%) or Somerset County (23%), which are home to large South Asian communities in NJ [4]. The remaining participants came from 4 other New Jersey counties (county names not provided to protect privacy of the participants due to small frequencies). These align with the concentration of NJ's South Asian population around Middlesex County and its adjacent counties–Middlesex County is the largest home and the adjoining Somerset County is the fourth largest home to NJ's South Asian population. Three of the 7 community-based organizations that disseminated our study information and the community-based radio have Middlesex County as their headquarters, and one out of the 7 community-based organizations operates from Somerset County.

## Lessons learned

**Leveraging broadcast media to strengthen community partnerships and credibility.**
Although the radio script includes the study's contact phone number and email, initially individuals reached out to the community radio to enquire about the study. The community radio station encouraged them to contact the study staff, and interested individuals subsequently reached out to us. This worked very well. We were also keen to build stronger connections with the radio listeners. Therefore, to help us build better rapport with the listeners, the community radio station invited us to participate in a talk show to describe our breast cancer research program and the SABCa study and summarize current advances in breast cancer prevention, early detection, and treatment. This was an opportunity for listeners to hear directly from us. Subsequently, participants started contacting us directly. Some participants also remarked that they found our research efforts to be very meaningful and that hearing the talk show helped them better understand the credentials and scientific credibility of the study investigators and the importance of this research.

**Handling fraudulent responses to study announcement.** In the study flyer, we provided email and phone numbers for interested individuals to contact the study staff. As an additional option, we also included a link to an IRB-approved online survey page, where interested individuals could provide their phone and email contact information so that study staff could contact them. This appeared to be a good plan at the outset. However, we encountered considerable challenges with this online survey shortly after community organizations disseminated the study flyer via their listserv and social media. We received 88 responses in this online survey. Of these, 85 responses were recorded on a single day, each only 2 seconds apart. The phone numbers provided for these responses differed by just one digit. The email addresses of all the responses had identical patterns: a few alphabets, followed by two digits, and @gmail.com. All the IP addresses were from CA. Due to these patterns, we concluded that these are fraudulent responses. We closed this survey page and decided to not use this further. In all our subsequent dissemination of study information through the community radio and Rutgers Co-op, we asked interested individuals to contact us via phone or email.

**Leveraging male champions of women's health.** Since the radio script for our study addressed South Asian American women, we expected only interested women to reach out to us. One South Asian American male heard the study promotion through the radio. This male reached out to us to learn about the study and gave the study's contact information to their spouse, encouraging the spouse to participate. Another South Asian American male heard the study through a radio talk show and encouraged their spouse to directly contact the study staff

and participate. Both spouses enrolled in the study. This highlighted the significant role of family dynamics in South Asian American communities and the potential for leveraging men as champions of women's health. Therefore, when we presented our study at an online community health program organized by Rutgers Co-op, we encouraged both women and men to join and asked everyone, not solely women, to share the study information with their networks.

**Assisting participants with technology challenges.** Although people were making considerable use of online resources during the COVID-19 pandemic, some study participants had difficulties navigating the online technology tools to join a virtual meeting room via Rutgers-approved Zoom® platform or provide electronic informed consent via Rutgers-approved Docusign®. Therefore, study staff prepared training modules for the participants, communicated with them via telephone to build rapport, and guided them with using these technology tools. Study staff were also flexible and offered alternative approaches such as allowing participants to submit an image of a signed paper copy of the informed consent. Being available to assist participants, empathizing with the comfort levels of individuals in using technology tools, and offering them alternative strategies were critical for retaining these participants in our study.

**Dialogues with community as a facilitator of trust and research participation.** Our participation in the community radio station's talk show and presentation at an online community health program organized by Rutgers Co-op allowed the audience to hear directly from us. The community radio station and Rutgers Co-op were our cultural research brokers, who were trusted by NJ's South Asian American community. These cultural research brokers introduced us to their audience by mentioning our credentials, including our advanced degrees, experience in the field, and prior research works. Such credentials are held in high regard by the South Asian American community. Together, these contributed to individuals building trust in our research and participating in our study. This trust also contributed to snowball sampling. Some participants became our advocates by enthusiastically sharing our study information with their acquaintances, encouraging them to participate.

At the time of enrollment, some participants informed us that they are keen for their young daughters and other family members to learn about breast cancer and asked us for resources. We directed them to Rutgers CINJ's Breast Cancer Resource Center [53]. In an effort to sustain our dialogues with the community, to continue to build trust, and to contribute to advancing the health of South Asian American women, the PI wrote an introductory article about breast cancer in a community organization's newsletter during October 2022 [54] and recruited a South Asian American woman to the CINJ Community Cancer Action Board.

## Discussion

In this paper, we described the approaches we took to recruit participants for the SABCa study from NJ's South Asian American community during the COVID-19 pandemic and summarized the lessons learned from our recruitment experiences. To our knowledge, this is the first study on recruitment of South Asian American women through community partnerships during the COVID-19 pandemic. We had considerable difficulty recruiting participants through community organizations since the pandemic-related needs of the communities placed significant demands on their time. We also received fraudulent responses to our recruitment efforts through a survey where interested individuals could provide their contact information for study staff to reach them. Participants' trust in the community radio as a credible source of information and their trust in the staff of online community health programs led by Rutgers Co-op led to successful recruitment. The community radio and Rutgers Co-op were credible

cultural brokers who helped the community understand the credentials and scientific credibility of the study staff and the importance the study. The overall cooperation rate was 88% and the rates corresponding to recruitment through the radio and Rutgers Co-op's networks exceeded 80%.

In our study, one (4.5%) out of the 22 total participants was recruited from community organizations. This participant was recruited after contacting 12 community organizations. Other studies have reported low participation of community organizations and low recruitments from community organizations during the COVID-19 pandemic. A study of Vietnamese Americans contacted 320 community organizations, 28 (8.75%) agreed to announce the study, and 68 participants were recruited i.e., 5 community organizations were contacted to recruit 1 participant [12]. In another study, 422 community organizations were contacted to recruit Korean Americans, 72 (17%) announced the study, and 13 participants were recruited i.e., 32 organizations were contacted to recruit 1 participant [55]. Due to low participation, Vu and colleagues recommend that studies of Asian Americans must contact a large number of community organizations to even achieve a modest sample size [12]. In a survey of researchers funded by the U.S. National Institute of Health to conduct clinical trials through community-based recruitment, respondents reported using online recruitment as one of the methods to navigate COVID-era recruitment challenges [56]. However, a study from Canada on mothers who have experienced domestic abuse and a UK-based study of South Asian women who had experienced gender-related violence found that collaborative relationships with community partners or gatekeepers is crucial for successful recruitment even when pursuing online recruitment [57,58]. The pandemic placed considerable strains on NJ's South Asian American community organizations, which may explain only one participant recruited through community organizations in our study.

However, tenacity is critical for achieving recruitment goals. To this end, Louw and colleagues recommend implementing innovative and creative approaches that involve building genuine and trustworthy conversations and collaborations with the community and maintaining that trust and relationships [59]. To build such connections with the community, we partnered with a community radio station catering to South Asian Americans and collaborated with Rutgers Co-op to engage with their community-based online health program and to reach their community networks. Through these experiences we learned the value of being prepared to pivot to alternative recruitment strategies when the pandemic precluded community organizations from devoting time and effort to assist with recruitment for research studies.

Our study's participants had lived in the US between 5 to 51 years (average = 26 years). All 22 participants were born in a South Asian country and are, thus, immigrants. The history of immigration of South Asians to the US dates to the early 1700s [60]. However, the growth of South Asian population in the U.S started increasing only after the Immigration and Nationality Act of 1965 [60,61]. A large number of immigrants came to the US seeking higher education or as working professionals with advanced degrees, including information technology (IT) professionals, and to meet the needs of the American economy of professionals and technology [60,62,63]. Several came as blue-collar workers and had less English fluency and less education than the professional immigrants [64,65], or came as families of students and professionals. Thus, the South Asian diaspora includes the upwardly mobile professional class and the blue-collar working class, and is diverse in education attainment and language proficiency, amongst other characteristics [60].

In our study, 14 participants have lived in the U.S for more than 20 years and, thus, belong to the first or second wave of immigrants, and the remaining 8 belong to the recent wave. The high proportion of master's or professional degrees and the use of English language (equal to

more than the use of South Asian languages) to speak with friends and as a preferred language for entertainment may reflect the typical demographics of South Asian immigrants who arrived as professionals or students. This suggests that additional strategies, over and above partnering with the community radio station and community-focused health programs, are needed to recruit South Asian Americans participants from other demographic subgroups. Indeed, a socio-behavioral study from Mozambique noted that engagement with community organizations is critical for recruiting participants during the COVID-19 pandemic, and that the use of alternative engagement approaches such as the radio or online platforms may vary among individuals [63]. A study of Muslim immigrant women and Arabic-speaking older adults from Canada used digital platforms to collect data during the COVID-19 pandemic but encountered difficulties engaging with those having low digital literacy [36], a concern also expressed by other studies [66,67]. A UK study of South Asian women facing gender violence, which recruited participants using Facebook and was conducted before the COVID-19 pandemic, also echoed the vital role of working closely with community organizations to ensure that 'invisible groups' are not created comprising those who do not prefer technology [58]. We began our study by partnering with a broad range of community organizations. However, the COVID-19 pandemic put considerable strain on their time and resources, limiting our ability to partner with a wide range of organizations. In the future, we must be mindful of such issues during times of crises, which could lead to recruitment from limited demographic subgroups.

A previous study reported paternalistic and shared family decisions and altruism to be important facilitators of recruitment and retention of South Asian Americans in clinical trials [68]. Indeed, two women in our study were encouraged by their male spouses to participate and some women told us that they decided to participate because they found our study to be meaningful. Prior studies identified lack of time and cost of participation, and logistical challenges including childcare and transportation as key barriers to research participation [5,6,69,70]. Since our study was conducted via a Rutgers-approved online Zoom® platform, there was no issue of transportation for the study participants. The COVID-19 pandemic exposed the dependence of world economies on the invisible and unpaid care labor of women [71]. Lack of time due to childcare and other family care may have been barriers for some women to enroll in our study even though our study did not require travel. Another barrier could have been technology challenges. Although we assisted participants with technology for joining the virtual Zoom® room and for providing electronic informed consent, some women may not have participated in the study due to technology concerns at the outset.

Several participants (45%) heard about the study through the community radio. Studies from the pre-COVID-19 era have recommended radio ads and live radio segments such as talk shows to recruit U.S minority populations, but a study of diabetes and obesity in South Asians from the UK found that disseminating study information through the radio was not successful for enrolling participants [72,73]. A study of recruitment preferences of various Asian American subgroups conducted during the COVID-19 pandemic in CA between February and May 2021 found that preference for hearing about a study through radio or television versus a study flyer varied across Asian American subgroups–Vietnamese Americans preferred ethnic radio or television, Japanese Americans preferred flyers, and Asian Indian Americans had no preference [74]. However, in our study conducted in NJ where recruitment was done between March 2021 and June 2022, only one participant was recruited by disseminating study flyers via email or social media through community organizations. The effectiveness of the community radio in our study may reflect differences between South Asians in CA and NJ or differences in the recruitment period.

Previous studies have found several facilitators for successful recruitment and retention of South Asian women in research, including cultural alignment where study staff are also

women of South Asian origin, cultural research brokers who possess advanced degrees, providing incentives, and altruism [5,69,70,75]. Cultural alignment is also an important facilitator for recruitment and retention in other Asian American cultures. For example, a study of Chinese Americans found integrating cultural values into study materials and messaging to be important strategies for maximizing engagement and retention [76]. In our study, in addition to cultural concordance of the study flyer and radio ads, 3 of this paper's authors (JMS, TD, and SM) represent diverse South Asian origins and have advanced degrees. Further, one of the authors (SM), who is affiliated with Rutgers Co-op, has long-standing ties with NJ's South Asian American community and conducts community health education programs. Two of the paper's authors (JMS and SM) also have long-standing connections with various South Asian cultural and religious organizations in NJ. Some participants perceived their participation to have health benefits for the broader community. We offered $25 electronic gift card for enrolling in the study and completing the structured survey. Systematic literature reviews have shown that these facilitators to minority research participation are also shared by African Americans, Latinos, and Filipino Americans [77] and by participants in Australia, Canada, South Africa, the UK, and other countries [78].

Studies from Europe [79,80], the UK and Africa [81,82], India [83], and the U.S [84] have found some participants to have difficulties using digital technology. In our study, some participants experienced difficulties connecting to the virtual room and in using the electronic platform to provide informed consent. One of the authors (TD) prepared a tutorial and guided the participants with the use of technology.

## Limitations

Our findings should be interpreted considering the following limitations. We conducted our study in English and met participants in a virtual room. The COVID-19 pandemic in the U.S and the delta variant wave in South Asia may have affected the lives and well-being of NJ's South Asian Americans in numerous ways, impacting their ability to participate in our research even if they were proficient in English and in the use of digital technology. We did not collect data on the COVID-19-related well-being of our study participants to determine whether our study consisted only of those individuals who were less affected by the pandemic.

English proficiency of South Asian Americans is 77% for Asian Indian Americans and 71% for Pakistani Americans, who constitute the largest South Asians in NJ [85,86]. Individuals who preferred using their native South Asian language instead of English and those not comfortable using digital technology to join virtual rooms would not have participated in our study. Due to pandemic-related budgetary limitations, we were unable to translate study material to multiple South Asian languages and recruit multi-lingual staff. However, with the pandemic now officially declared over, future research can consider conducting the study in multiple languages and as in-person interactions without requiring the use of digital technology.

Our study participants had high educational attainment. Snowball sampling could have led to participants having similar demographics. South Asian Americans are a diverse population and have varying sociodemographic status, including less than high school education; nearly 10% of this population lives under poverty and around 12% are undocumented [87]. Alternative strategies such as partnering with cultural gatekeepers should be pursued in future studies to engage this subpopulation who may otherwise remain invisible for research.

Facebook is used widely by South Asian Americans [88]. However, we did not place targeted Facebook ads due to infodemic concerns and budgetary limitations. Instead, we encouraged community organizations to post our study flyer in their community-focused Facebook

groups. However, only those community members who engage with these Facebook groups would see the study information. Future research can pursue targeted Facebook ads and compare the resulting recruitment and retention with those based on posting in community-focused Facebook groups to examine the benefits of targeted ads and the characteristics of participants recruited through this approach.

## Conclusion

In this paper, we have described our experiences and lessons learned in community-based recruitment of South Asian Americans to the SABCa study in NJ during the COVID-19 pandemic. We began the study by partnering with various community organizations. However, we had to pivot to alternative strategies such as advertising through a community radio station and leveraging Rutgers Co-op's community network and online health program. Using culturally tailored outreach approaches, trust in the community radio and Rutgers Co-op, position of the staff of these organizations as credible cultural brokers in the community, and family dynamics were among the facilitators for recruitment and retention of participants. Our study can inform strategies for recruiting understudied populations to research studies even beyond the pandemic. Future studies can consider additional strategies, over and above partnering with the community radio and community-focused health programs, to recruit South Asian American participants from more diverse demographic subgroups.

## Supporting information

**S1 Fig. The South Asian breast cancer (SABCa) study flyer.** The watermark shows heaps of spices commonly used South Asian cuisines. The yellow colors represent turmeric, a commonly used spice. The red color in the border and horizontal lines represent powdered chili, a commonly used spice, and also represent henna and vermillion, which are commonly used for cosmetic needs in the South Asian culture. The phrase "VISIT: SAQHE Group Discussion" gave a link to a Rutgers-approved Qualtrics survey page where interested individuals could provide their contact information (phone number and email address) so that study staff could contact them.
(TIF)

**S2 Fig. Radio script.** This IRB-approved script was used to disseminate the study information through the community radio.
(TIF)

## Acknowledgments

The authors thank the study participants, the community-based organizations, and the community radio.

## Author Contributions

**Conceptualization:** Jaya M. Satagopan, Elisa V. Bandera, Anita Y. Kinney.

**Data curation:** Jaya M. Satagopan, Tina Dharamdasani.

**Formal analysis:** Jaya M. Satagopan, Tina Dharamdasani.

**Funding acquisition:** Jaya M. Satagopan, Anita Y. Kinney.

**Investigation:** Jaya M. Satagopan.

**Methodology:** Jaya M. Satagopan.

**Project administration:** Jaya M. Satagopan, Tina Dharamdasani, Shailja Mathur.

**Resources:** Jaya M. Satagopan, Anita Y. Kinney.

**Software:** Jaya M. Satagopan, Tina Dharamdasani.

**Supervision:** Jaya M. Satagopan.

**Writing – original draft:** Jaya M. Satagopan, Tina Dharamdasani.

**Writing – review & editing:** Jaya M. Satagopan, Tina Dharamdasani, Shailja Mathur, Racquel E. Kohler, Elisa V. Bandera, Anita Y. Kinney.

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
