## [Decision Letter · Decision Letter 0]

9 May 2023

PONE-D-23-05763Lessons learned from community-based recruitment for the south Asian breast cancer study during the COVID-19 pandemicPLOS ONE

Dear Dr. Satagopan,

Thank you for submitting your manuscript to PLOS ONE. After careful consideration, we feel that it has merit but does not fully meet PLOS ONE’s publication criteria as it currently stands. Therefore, we invite you to submit a revised version of the manuscript that addresses the points raised during the review process.

We look forward to receiving your revised manuscript.

Kind regards,

Erika Bonnevie

Academic Editor

PLOS ONE

Reviewers' comments:

Reviewer's Responses to Questions

**Comments to the Author**

1. Is the manuscript technically sound, and do the data support the conclusions?

Reviewer #1: Partly

Reviewer #2: Partly

2. Has the statistical analysis been performed appropriately and rigorously? 

Reviewer #1: N/A

Reviewer #2: I Don't Know

3. Have the authors made all data underlying the findings in their manuscript fully available?

Reviewer #1: No

Reviewer #2: Yes

4. Is the manuscript presented in an intelligible fashion and written in standard English?

Reviewer #1: Yes

Reviewer #2: Yes

5. Review Comments to the Author

Reviewer #1: Title Main title:

• There is no indication of where the study was conducted; the way it is stated it would imply that the study was conducted in South Asia

• Authors should modify the title to give idea of the study location

Abstract Methods: Authors should state the category of people recruited and where (location) the recruitment covered

• The study design, number of participants should also be mentioned

Results:

• On line 30, authors stated that they wanted to “gain insights into breast cancer-related health attitudes of South Asian American”, however there is no single figure in the result regarding breast cancer health related attitudes of the women; the result mainly focused on awareness of the women on the study and their sources of information.

• Authors should re-write the result section in line with the aim of the study, otherwise they will need to modify the study title in line with what they presented (i.e. awareness on the breast cancer study on sources of information)

Conclusion: Does not appear to be based on your objectives; kindly modify it to capture your objectives

Key words:

• Authors did not provide any keyword; kindly provide at least 3-5 keywords

Introduction 1. There is need for authors to fragment the 1st paragraph (line 54-85) into different paragraphs as new ideas are introduced

2. Authors did not provide information regarding breast cancer-related health attitudes attitude of women as reported in other studies; this will give insight into the burden of the problem, since it is part of the objectives of the study

3. Rationale/aim of the study should be taken towards the end of the introduction (preferable last paragraph)

4. There is need to show the various nuanced approaches used in different parts of the globe as reported in similar studies

Methodology Study area:

• Authors should begin this section with description of the study area

Study population:

• Authors need to state the study population before describing the eligibility criteria

•

Study design:

• Kindly state the study design used

Instrument of data collection:

• Authors did not describe the instrument of data collection; they should kindly state and describe the instrument used for the data collection

Data source and collection:

• Authors said they used FGD, however, there is no description of the FGD that was conducted, what categories of participants constituted each focus group, were the groups homogenous? Etc Authors need to provide more information on these issues

Recruitment method:

• In line 133, the authors said snowball sampling was used to recruit participants, however, in line 134/135 they said passive sampling….was used; there is need for clarification.

• If snowball sampling was used, what was the rationale behind that?

Fig 1. Timeline of community engagement strategies.

• No figure shown in the manuscript; what authors presented as figure (last page) looks more like a table rather than figure

Community organizations

• The detail description of the organizations given by authors is not necessary as they are not the focus of the study

Community-focused health education events

• This sub-heading gives the impression that the study was an intervention study (educational intervention); authors may need to concentrate on describing methods related to attaining the objectives of their study

Sample size:

• Authors said they sought to recruit 18 to 32 eligible participants to conduct 3 to 4 focus groups; authors should state the actual number of participants recruited and the number of FGD groups since the study has been conducted already

• For a qualitative research that adopted FGD as the only method of data collection, optimal number of FGDs to be conducted is determined by data saturation rather than a predetermined number set by the researcher.

Statistical analysis:

• The description given by authors gives the impression that quantitative data analysis was done. If that was the case, the sample size is too small and there would be need to show the formula that was used to calculate the sample size

• Authors did not describe how the FGD was analysed; was it narrative synthesis or it was using software such Nvivo etc?

•

Results Result:

• Most part of the result concentrated on describing participants’ characteristics rather than providing information on the objectives of the study

• No result for the FGD?

Lessons learned

• From the list of lessons learned, it is clear the research focused communication/community engagement strategies rather than breast cancer-related attitude of women

Discussion

Conclusion/recommendations • Authors should provide conclusion and recommendations

Abbreviations Provide meanings for some of the abbreviations used in the main text

Bibliography/References Reference list (No. 2, 25): Correct authors’ name in line with Vancouver referencing style

Reference list (No. 6,16,18,28,33): update page number (or provide web address/access date)

Reference list (No. 24): Provide place of publication

Reference list (No. 19,20,21,22,26): provide date article was accessed from the web source

Reference list (No. 38): web address provided is not accessible (remark: “page not found”)

Others Nil

Final Note The manuscript may be considered for publication after effecting corrections.

Reviewer #2: General

Do a review for grammar/ spelling. I found a few errors throughout.

The sentences that start out with brackets generally dont really need the brackets. I would remove the brackets and make sure the sentence flows well within the paragraph.

Introduction

Is there a rationale for establishing the South Asian Breast Cancer (SABCa) study in New Jersey, aside from the Rutgers location? For example, is there a large South Asian community in New Jersey? The introduction could use more information about the South Asian community in the state, especially given that it focuses so heavily on lessons learned.

In Line 78, the way it is written, it appears that “In-person meetings of community organizations were suspended” is referencing the SABCa study. I would reorganize so the introduction clearly states the problem, the population that is being studied (South Asians in New Jersey), and how community organizations/ research organizations managed the COVID pandemic

What is the reason for focusing just on Objective 1? Given the small sample size for this current paper, and the fact that Components ⅔ also were qualitative approaches, there could be good lessons learned across all components that would be suitable for a manuscript. I would rethink why the paper only focuses on Objective 1. It would be a far stronger paper if it referenced the entire project.

Methods

The authors do a thorough job of describing the methods that they eventually adopted, but there is less information on what the initial methods were, and how they decided to shift them. Given that this is an article on lessons learned, I would like to see more information in the methods on how they changed approaches, and why. This one of my largest issues with the paper.

There is a need for more information on how outreach efforts were tailored to reach South Asians specifically. For example, was outreach done in just English?

Did the authors think about doing Facebook ads or doing social media outreach, outside of the cultural, religious, health screening, and health service organizations? If not, why? Does this group not use social media? This seems like an easy way to advertise to a wide group during a time when people were spending a lot of time on the internet.

Results

The results focus mostly on demographics and aren’t particularly compelling, particularly given the small sample size. I understand that the focus of this paper isn’t to review the results of the study, but is more a reflection on the process, but as-is, the results seem like they can be removed entirely without affecting much. I would think through what other information you can provide about the study or recruitment methods to give the results section a more interesting angle.

Lessons Learned

There is a lot of information here that I would like to see in the methods. For example, lines 297-305 should be put in the methods section. This is similar throughout this section.

Lines 305-306 - I would like to see a lot more information on this piece - specifically culturally aligned approaches. That is the crux of this paper, but I dont see a lot of information on this aspect, except for here.

In general, the conclusions section feels more like a methods section. The conclusions section should reflect on the types of research out there on community recruitment, trusted messengers, etc.

Conclusions

This is where authors bring in some outside research, but it’s buried at the end of the lessons learned section, which feels more like a methods section to me.

The conclusions section smushes too many ideas into a paragraph, when they should be separated, and discussed as the lessons learned. For example, the paragraph starting on line 395 has the multiple competing ideas about recruitment in one paragraph - when the entire article is about recruitment, and these should be broken up into multiple different paragraphs, with a clear “lesson learned,” reflection on the authors’ own work, and what has already been published. There are opportunities to think broadly about health communications and how they have changed since COVID (which is purportedly the point of the article), but I dont get much of that in the conclusions, or throughout.

What was the messaging that was used to recruit people? This sentence in the conclusions brings up a good point that is not address in the methods. Having a sample image of recruitment materials would be helpful to show the specific messages that were used, and how they were tailored to reach South Asian audiences: “willingness to improve health and contribute towards scientific knowledge and society at large encouraged South Asian participation in research”

Limitations

The paper needs a stronger limitations section. There are general references to limitations sprinkled throughout, but it needs a condensed paragraph focusing just on the limitations. There are several that require addressing. The biggest ones that I can think of: 1) Language - it seems like everything was done in English, so you are missing people who don’t speak English and are likely to be more vulnerable. In line 415-416, you make it seem like this was something that happened to you - when in fact, this was eligibility criteria. 2) Educational background - The biggest issue to me is that 68% of the participants had master’s, professional, or doctoral degree - that seems way out of proportion for the actual study population that you are trying to reach.

6. PLOS authors have the option to publish the peer review history of their article (what does this mean?). If published, this will include your full peer review and any attached files.

Reviewer #1: **Yes: **Dr Habibullah Adamu

Reviewer #2: No

---

## [Author Response · Author response to Decision Letter 0]

4 Oct 2023

Response to reviewers is attached as a word document. It is copied and pasted below as well. 

Responses to comments from Reviewer 1

Title 

Main title:

• There is no indication of where the study was conducted; the way it is stated it would imply that the study was conducted in South Asia

• Authors should modify the title to give idea of the study location

We conducted this study in New Jersey, USA. We have now modified the title to indicate the study location. The revised title of our paper is: “Experiences and lessons learned from community-based partnerships in recruitment for the South Asian breast cancer (SABCa) study in New Jersey during the COVID-19 pandemic”. 

Abstract 

Methods: Authors should state the category of people recruited and where (location)

the recruitment covered

• The study design, number of participants should also be mentioned

We have revised the abstract as recommended. 

Results:

• On line 30, authors stated that they wanted to “gain insights into breast cancer-related health attitudes of South Asian American”, however there is no single figure in the result regarding breast cancer health related attitudes of the women; the result mainly focused on awareness of the women on the study and their sources of information.

There are considerable data gaps and knowledge gaps regarding breast cancer-related health of South Asian American women. To fill these gaps, we established the South Asian Breast Cancer study in New Jersey during early 2020 to gain insights into breast cancer-related health attitudes of South Asian American women by partnering with the community. In this paper, we focus only on study recruitment (and not on the breast cancer-related health attitudes) for the South Asian Breast Cancer study in New Jersey using community partnerships. Therefore, we describe the approaches we used, and lessons learned during such recruitment. To clarify this, we have now included a new paragraph (second paragraph) on page 6 (lines 113-120) in the Introduction section of the revised paper. 

Authors should re-write the result section in line with the aim of the study, otherwise they

will need to modify the study title in line with what they presented (i.e. awareness on the

breast cancer study on sources of information)

The goal of this paper is to describe our experience and lessons learned during community-based recruitment for the South Asian Breast Cancer study in New Jersey during the COVID-19 pandemic. Therefore, our revised title and the results section are in line with the aim of our paper. 

Conclusion: 

Does not appear to be based on your objectives; kindly modify it to capture your

Objectives

The goal of this paper is to describe our experiences and lessons learned during community-based recruitment for the South Asian Breast Cancer study in New Jersey during the COVID-19 pandemic. The abstract conclusion of our revised paper is in line with this aim.

Key words:

• Authors did not provide any keyword; kindly provide at least 3-5 keywords

The journal’s instruction for submission does not ask us to provide keywords. Hence, we did not include keywords in our previous submission. We have provided a few keywords in this revision. 

Introduction 

1. There is need for authors to fragment the 1st paragraph (line 54-85) into different paragraphs as new ideas are introduced

We have now fragmented the introduction section into different paragraphs as suggested. 

2. Authors did not provide information regarding breast cancer-related health attitudes attitude of women as reported in other studies; this will give insight into the burden of the problem, since it is part of the objectives of the study

In the revised paper, we have referred to other research works about breast cancer-related health attitudes of women in the Introduction section. We summarize the key studies in the paragraph “Breast cancer is a growing concern …” in the Introduction section. Several studies have examined motivators and barriers to breast cancer screening practices of South Asian women from the UK and Canada. Corresponding studies of South Asian American women remain limited.

3. Rationale/aim of the study should be taken towards the end of the introduction (preferable last paragraph)

We have now moved the objective to the end of the introduction section. 

4. There is need to show the various nuanced approaches used in different parts of the globe as reported in similar studies

We have now briefly discussed approaches used for recruitment of South Asian Americans in similar studies in the Introduction section (lines 134-136). In the Discussion section, we refer to publications that have used various approaches to recruit individuals from diverse race/ethnicity in pages 24 and 25 (lines 504-522) and in the first paragraph of page 28 (lines 585-598). 

Methods

Authors should begin this section with description of the study area

We now provide the study area at the beginning of the Methods section in the line “Recruitment was done locally …”. We have also included a paragraph in the Introduction section to summarize a rationale for conducting the study in Rutgers University – please see paragraph “A vast majority of …”. 

Authors need to state the study population before describing the eligibility criteria

We now provide the study population as the first sentence of the Methods section before describing the eligibility criteria in this revision. 

Kindly state the study design used

We now provide a short subsection called Study Design in the Methods section (page 8, lines 169-170), where we indicate that we used a cross-sectional design and collected data from participants at a single time point. 

Authors did not describe the instrument of data collection; they should kindly state and

describe the instrument used for the data collection

We have provided a subsection called “Participant survey” under the Methods section, where we describe the instrument of data collection (pages 15 and 16, lines 342-349). Briefly, we administered a structured survey hosted on a Rutgers-approved Qualtrics® platform. We administered the survey through a Rutgers-approved virtual platform at a mutually agreed date and time between the participant and the research staff. 

Authors said they used FGD, however, there is no description of the FGD that was

conducted, what categories of participants constituted each focus group, were the groups

homogenous? Etc Authors need to provide more information on these issues

In this paper, we focus only on the recruitment for the South Asian Breast Cancer study in New Jersey using community partnerships. Therefore, the objective of this paper is to describe the recruitment approaches that we used, and lessons learned during such recruitment. Qualitative analyses, including composition of focus groups will be the main focus of another subsequent paper. This is stated in page 6 (lines 113-120) in the Introduction section. 

In line 133, the authors said snowball sampling was used to recruit participants, however, in line 134/135 they said passive sampling….was used; there is need for clarification.

We used snowball sampling strategy for our South Asian Breast Cancer Study (SABCa) study recruitment. Our recruitment method using the snowball sampling strategy was passive, meaning that interested individuals contacted the study personnel and were then enrolled if eligible. In particular, the study personnel did not actively contact individuals in the community for their interest in participating in the SABCa study. To further clarify, we have removed the words “passive sampling” in the revised manuscript. 

If snowball sampling was used, what was the rationale behind that?

We used snowball sampling strategy for our study recruitment. Previous studies have effectively recruited South Asian participants using snowball sampling strategy and recommended using this strategy for recruitment of hard to reach populations such as South Asians (Kanaya et al., 2019, Journal of Clinical and Translational Science, 3:97-104). In a study of mobile health usage among Vietnamese Americans, Vu et al (2021, PLOS One, 16(8): e0256074) found snowball sampling to be a successful recruitment method. We have added these citations to provide rationale for snowball sampling for our study in the revised manuscript. 

Fig 1. Timeline of community engagement strategies.

• No figure shown in the manuscript; what authors presented as figure (last page) looks more like a table rather than figure

We changed “Figure 1” to ‘Table 1” in the revised manuscript. 

Community organizations

• The detail description of the organizations given by authors is not necessary as they are not the focus of the study

We have made considerable modifications to the manuscript to incorporate this suggestion from this reviewer. Further, since the goal of this paper is to describe various community-based recruitment approaches and lessons learned from those approaches for the SABCa study, we have included description of community organizations. We have summarized our recruitment approaches under subheadings “South Asian community-based organizations”, “Broadcast media (radio)”, “Rutgers-facilitated community outreach”, and “Snowball sampling”. 

Community-focused health education events

• This sub-heading gives the impression that the study was an intervention study (educational intervention); authors may need to concentrate on describing methods related to attaining the objectives of their study

We have edited this sub-heading. In the revised manuscript, we have a subheading “Rutgers-facilitated community outreach” where we clarify that our study information was disseminated through community-focused health education events of Rutgers Co-op and through community networks of Rutgers Co-op. 

Sample size:

• Authors said they sought to recruit 18 to 32 eligible participants to conduct 3 to 4 focus

groups; authors should state the actual number of participants recruited and the number of FGD groups since the study has been conducted already

In qualitative studies that use focus groups, data saturation is certainly the gold standard approach for determining the number of focus groups and, thus, the total number of individuals in a study. By definition, saturation can be determined only during or after data analysis. Analysis of the focus group transcripts are ongoing at the time of writing this revised manuscript and response to reviewers. These ongoing analyses indicate that saturation was reached with the sample size of 22 participants enrolled to our study. 

For a qualitative research that adopted FGD as the only method of data collection, optimal number of FGDs to be conducted is determined by data saturation rather than a predetermined number set by the researcher.

We agree. Our target sample size was motivated by the qualitative methods literature. We include this in page 9 of the revised manuscript. 

The description given by authors gives the impression that quantitative data analysis was done. If that was the case, the sample size is too small and there would be need to show the formula that was used to calculate the sample size

Qualitative analysis of focus groups and interviews, and recruitment of healthcare professionals and South Asian women with breast cancer are not the focus of this paper. In this paper, we focus only on the recruitment of women without a prior diagnosis of breast cancer for the South Asian Breast Cancer study focus groups in New Jersey using community partnerships. We have stated this in page 6 (lines 113-120) in the Introduction section. 

Authors did not describe how the FGD was analysed; was it narrative synthesis or it was

using software such Nvivo etc?

This was addressed as a prior reply to the reviewer’s comment. The objective of this paper is to describe the approaches used and lessons learned during community-based recruitment of South Asian Americans during the COVID-19 pandemic. Qualitative analyses of the focus groups will be the main focus of another subsequent paper. We have described this in page 6 (lines 113-120) in the Introduction section. 

Results:

• Most part of the result concentrated on describing participants’ characteristics rather than providing information on the objectives of the study

• No result for the FGD?

As mentioned in the responses to prior comments, the objective of this paper is to describe our experiences and lessons learned during community-based recruitment of South Americans during the COVID-19 pandemic. We recruited these participants to conduct focus group discussions. However, in this paper, we focus on describing recruitment results. Analysis and results of focus group discussions will be reported in a future paper. We explain the focus of our paper in page 6 (lines 113-120) in the Introduction section. 

Lessons learned

• From the list of lessons learned, it is clear the research focused communication / community engagement strategies rather than breast cancer-related attitude of women.

The primary goal of the SABCa study is to gain insights into breast cancer-related attitude of South Asian American women. To this end, we had to recruit participants by partnering with the South Asian American community in New Jersey. As described above, in this paper, we describe our experiences and lessons learned during community-based recruitment of South Americans during the COVID-19 pandemic. This paper contributes to a scant body of literature on this topic and lessons learned that we describe may be useful to other researchers. Analysis and results of focus group discussions will be reported in a future paper. Hence, our lessons learned relate to our community engagement strategies. 

Discussion

Conclusion / recommendations. Authors should provide conclusion and recommendations. 

We have included a Conclusion section in the revised paper. 

Abbreviations. Provide meanings for some of the abbreviations in the main text. 

We elaborate our abbreviation when it used first. The following abbreviations are described: 

United States (US) in page 4

New Jersey (NJ) in page 4

United Kingdom (UK) in page 5

The South Asian Breast Cancer (SABCa) study in page 5

California (CA) in page 6

Rutgers Co-operative (Rutgers Co-op) in page 10

Rutgers Cancer Institute of New Jersey Community Outreach and Engagement (CINJ COE) in page 10

Standard Deviation (SD) in page 19

Bibliography / References 

References list (No. 2, 25): Correct authors’ name in line with Vancouver referencing style

We have reformatted the author’s name for reference number 2 [Hoeffel et al]. 

For reference 25 [reference 4 in the revised manuscript], the author is the organization “Jersey Promise”. Hence, we have retained the author name in the reference as before.

Reference list (No. 6, 16, 18, 28, 33): update page number (or provide web address / access date)

We checked the page numbers for these references and found that they are indeed as we have provided in our reference list. For example, for reference number 6 [which is reference number 13 in the revised manuscript], the page number given by the journal is indeed pkaa005. As suggested by the reviewer, we now provide web address and the most recently cited date. We also examined the citation format in another paper published in PLoS One, and used the same approach to format our references.

Reference list (No. 24): Provide place of publication

We have included the place of publication in the revised manuscript. 

Reference list (No. 19, 20, 21, 22, 26): provide date article was accessed from the web source

We have now provided our most recent date of accessing these articles. We have used the same format used in other papers published in PLoS One to include the date. 

Reference list (No. 38): web address provided is not accessible (remark: “page not found”)

We verified and found that are able to access this web page from New Jersey. We do not get “page not found” note. However, in the revised manuscript, we no longer cite this web address. 

Responses to comments from Reviewer 2

Introduction:

Is there a rationale for establishing the South Asian Breast Cancer (SABCa) study in New

Jersey, aside from the Rutgers location? For example, is there a large South Asian community in New Jersey? The introduction could use more information about the South Asian community in the state, especially given that it focuses so heavily on lessons learned.

Thank you for this comment. Yes, New Jersey has a large and growing South Asian population. South Asians are the largest Asian subgroup in New Jersey. A vast majority of New Jersey’s South Asians also live within reach of Rutgers University. We now provide these details in pgae 6 (lines 122-126) in the Introduction section of the revised paper. 

In Line 78, the way it is written, it appears that “In-person meetings of community

organizations were suspended” is referencing the SABCa study. I would reorganize so the introduction clearly states the problem, the population that is being studied (South Asians in New Jersey), and how community organizations/ research organizations managed the COVID Pandemic.

We have now edited this sentence in the Introduction of the revised paper. This sentence is now rewritten as “The South Asian community organizations in NJ suspended their in-person meetings and events due to lock downs and social distancing requirements”. We have also reorganized the Introduction section considerably as recommended by this reviewer. 

What is the reason for focusing just on Objective 1? Given the small sample size for this

current paper, and the fact that Components 2 / 3 also were qualitative approaches, there could be good lessons learned across all components that would be suitable for a manuscript. I would rethink why the paper only focuses on Objective 1. It would be a far stronger paper if it referenced the entire project.

Component 1 uses community-based recruitment methods, while component 2 recruits by partnering with the NJ State Cancer Registry and component 3 recruits healthcare professionals from Rutgers hospitals. Our study focuses on our experiences and lessons learned from community-based recruitment. Hence, we focus on Component 1. We have now described this in the Introduction section of the revised paper. Please see the paragraph “Therefore, we established …” 

Methods:

The authors do a thorough job of describing the methods that they eventually adopted, but there is less information on what the initial methods were, and how they decided to shift them. Given that this is an article on lessons learned, I would like to see more information in the methods on how they changed approaches, and why. This one of my largest issues with the paper.

We now provide these details in the Introduction section. Please see the paragraph “Several studies have recommended …” in the Introduction. If this study were to be conducted in a pre-pandemic era, recruitment methods would typically involve attending community-based events to disseminate information about the study and recruit participants. However, since community-based events were suspended due to the COVID-19 pandemic, we had to pivot to alternative strategies. One potential possibility was advertising through social media such as Facebook. However, the pandemic was also a period of infodemic. Therefore, we had to be wary of placing advertisements about our study on social media. Hence, we used the methods described in this paper – namely, leveraging broadcast media (radio) and Rutgers-facilitated community outreach. We have described these recruitment approaches in the Materials and Methods section of the revised paper. 

There is a need for more information on how outreach efforts were tailored to reach South Asians specifically. For example, was outreach done in just English?

Yes, the study outreach was done in English language only, since we did not have the resources to translate study materials to multiple South Asian languages, or to recruit research staff with multiple language skills. We have now added this information to the “Recruitment methods” subsection under the Materials and Methods section of the revised paper. Please see sentence “Study outreach efforts were done in English language only …”. 

Did the authors think about doing Facebook ads or doing social media outreach, outside of the cultural, religious, health screening, and health service organizations? If not, why? Does this group not use social media? This seems like an easy way to advertise to a wide group during a time when people were spending a lot of time on the internet.

No, we did not use social media for the SABCa study due to the misinformation that was being spread about science and health during the COVID-19 period. Therefore, we decided to not pursue recruitment by placing advertisements through social media. However, we did encourage the community-based organizations to share the study information through their social media groups. We also established a Qualtrics survey, where interested individuals could submit their contact info for us to contact them for the study, and we included a link to the Qualtrics survey in our study flyer. But we found that this page was being populated by bots and not bona fide individuals.

Results

The results focus mostly on demographics and aren’t particularly compelling, particularly given the small sample size. I understand that the focus of this paper isn’t to review the results of the study, but is more a reflection on the process, but as-is, the results seem like they can be removed entirely without affecting much. I would think through what other information you can provide about the study or recruitment methods to give the results section a more interesting angle.

We have made considerable revisions to the manuscript to address various comments from the two reviewers. We describe various community engagement strategies that we pursued and explain that we had to pivot to community-based recruitment by partnering with a community radio and Rutgers-facilitated community outreach program. Following these descriptions, we feel that it will be important to describe the demographic characteristics of the participants that we were able to recruit. To this end, we are retaining the results giving a summary of the participant characteristics and hope that this reviewer will support this. We have now moved the lessons learned section as a subsection of the Results section as a way to give a relevant angle since our paper is about experiences and lessons learned from our community-based recruitment efforts of South Asian Americans during the COVID-19 pandemic. 

Lessons Learned

There is a lot of information here that I would like to see in the methods. For example, lines 297-305 should be put in the methods section. This is similar throughout this section.

We have made considerable changes to the manuscript by moving many materials from the Lessons Learned section to the Materials and Methods section of the paper, as suggested by the reviewer. We have included these details within the “South Asian community-based organizations” and “Broadcast media (radio)” subsections of the Materials and Methods section. 

Lines 305-306 - I would like to see a lot more information on this piece - specifically

culturally aligned approaches. That is the crux of this paper, but I dont see a lot of information on this aspect, except for here.

Our culturally aligned approaches occur through 3 ways: (1) study flyer; (2) delivery of radio script; and (3) cultural concordance of study staff. 

In the revised paper, we have included a subsection called “Study flyer” within the Materials and Methods section, where we describe how our study flyer is culturally aligned with the South Asian culture. We have also included the study flyer as a supplementary material. 

We have included a paragraph “The radio script to promote …” in the “Broadcast media (radio)” subsection of the Materials and Methods section to describe how the radio script was delivered in a culturally aligned manner. The radio script is also included as a supplementary material. 

In the “Participant registration” subsection of the Materials and Methods section, we now highlight that the first 3 authors are women of South Asian origin and, thus, have cultural concordance with the study participants. 

In general, the conclusions section feels more like a methods section. The conclusions section should reflect on the types of research out there on community recruitment, trusted messengers, etc.

We have made substantial revisions to the Materials and Methods and Discussion sections. We have now included a Conclusion section, as suggested by the reviewer. 

Conclusions

This is where authors bring in some outside research, but it’s buried at the end of the lessons learned section, which feels more like a methods section to me.

To address this, we have made considerable revisions to the Materials and Methods section and Discussion section, and included a new Conclusion section. 

The conclusions section smushes too many ideas into a paragraph, when they should be

separated, and discussed as the lessons learned. For example, the paragraph starting on line 395 has the multiple competing ideas about recruitment in one paragraph - when the entire article is about recruitment, and these should be broken up into multiple different paragraphs, with a clear “lesson learned,” reflection on the authors’ own work, and what has already been published. There are opportunities to think broadly about health communications and how they have changed since COVID (which is purportedly the point of the article), but I dont get much of that in the conclusions, or throughout.

We have made several changes to our manuscript to address these helpful points. First, we moved several sentences or concepts from the previous Discussion section into the current Materials and Methods section. Second, we moved the “Lessons Learned” into the Results section of the revised manuscript. We also revised the “Lessons Learned” section to reflect on our own work. Finally, we rewrote our Discussion section to reflect more on our work. In rewriting the Discussion section, we used the paper by Vu et al (2021, PLOS One, 16(8): e0256074) as a template and role model for structuring our revised Discussion section. 

What was the messaging that was used to recruit people? This sentence in the conclusions brings up a good point that is not address in the methods. Having a sample image of recruitment materials would be helpful to show the specific messages that were used, and how they were tailored to reach South Asian audiences: “willingness to improve health and contribute towards scientific knowledge and society at large encouraged South Asian participation in research”

We now provide the study flyer and radio script as Supplementary Materials so share our study’s messaging with the readers of our paper. In the revised paper, we have also included how our participation in a radio talk show helped listeners to hear directly from us, which helped us build better connections with the community. We describe this in the first “Lessons learned” subsection within the Results section. 

Limitations

The paper needs a stronger limitations section. There are general references to limitations sprinkled throughout, but it needs a condensed paragraph focusing just on the limitations. There are several that require addressing. The biggest ones that I can think of: 1) Language – it seems like everything was done in English, so you are missing people who don’t speak English and are likely to be more vulnerable. In line 415-416, you make it seem like this was something that happened to you - when in fact, this was eligibility criteria. 2) Educational background - The biggest issue to me is that 68% of the participants had master’s, professional, or doctoral degree - that seems way out of proportion for the actual study population that you are trying to reach.

We have now included a Limitations section in our revised paper where we have collected the limitations sprinkled throughout the paper under one section. We have also highlighted that English is an eligibility criterion and that we did not advertise through Facebook, and list these as among the Limitations of the paper. We have also included the high educational background of the participants in the limitation and note that English language requirement as a potential reason for the high educational background. In addition, we also summarize a history of South Asians in the U.S and point to some aspects of the history in relation to the educational background. In the Limitation section, we now note the need to expand the study in the future by including South Asian languages and by partnering with diverse community organizations, especially now that we are in a post-pandemic time. 

Responses to Journal Comments:

We have carefully reviewed our entire manuscript to ensure that it meets PLOS ONE’s style requirements and file naming. 

In your Data Availability statement, you have not specified where the minimal data set

underlying the results described in your manuscript can be found.

We are including the following Data Availability statement as part of our online submission: 

The data that support the findings of this paper are available upon reasonable request and with the approval of Rutgers University Institutional Review Board (IRB Pro2020002217). The data are not publicly available as it contains information that could compromise the privacy of research participants. Given that participants are women of South Asian origin, and the dataset contains their county of residence and age, making the data available can risk the possibility of participants being identified. Please see more information here: https://policies.rutgers.edu/10011-currentpdf. Please contact the lead author (Jaya M. Satagopan; satagopj@sph.rutgers.edu) as well as Rutgers University’s Institutional Review Board (IRBOffice@research.rutgers.edu) with any request for data access. 

Please include your full ethics statement in the ‘Methods’ section of your manuscript file. In your statement, please include the full name of the IRB or ethics committee who approved or waived your study, as well as whether or not you obtained informed written or verbal consent. If consent was waived for your study, please include this information in your statement as well.

In the Materials and Methods section, we have included the full name of the IRB: “The Rutgers University Institutional Review Board (IRB) approved the study.” 

All study participants provided written informed consent through an electronic platform. We have included a sentence stating this: “All study participants provided electronic written informed consent through a Rutgers-approved secure Docusign® platform.”

---

## [Editor Report · Decision Letter 1]

27 Oct 2023

Experiences and lessons learned from community-engaged recruitment for the South Asian breast cancer study in New Jersey during the COVID-19 pandemic

PONE-D-23-05763R1

Dear Dr. Satagopan,

We’re pleased to inform you that your manuscript has been judged scientifically suitable for publication and will be formally accepted for publication once it meets all outstanding technical requirements.

Kind regards,

Erika Bonnevie

Academic Editor

PLOS ONE

Additional Editor Comments (optional):

Reviewers' comments:

Authors have addressed concerns and the paper is much stronger. I would suggest doing a general grammar review.

---

## [Editor Report · Acceptance letter]

5 Nov 2023

PONE-D-23-05763R1 

Experiences and lessons learned from community-engaged recruitment for the South Asian breast cancer study in New Jersey during the COVID-19 pandemic 

Dear Dr. Satagopan:

I'm pleased to inform you that your manuscript has been deemed suitable for publication in PLOS ONE. Congratulations! Your manuscript is now with our production department. 

Kind regards, 

on behalf of

Dr. Erika Bonnevie 

Academic Editor

PLOS ONE